# Isolated Peritoneal Metastasis of Prostate Cancer Presenting with Massive Ascites: A Case Report

Hee Ryeong Jang [1] , Kyoungyul Lee [2] and Kyu-Hyoung Lim [1,*]

1   Department of Internal Medicine, Kangwon National University Hospital, Kangwon National University School of Medicine, Chuncheon-si 24289, Gangwon-do, Korea; jangheeryeong@kangwon.ac.kr
2   Department of Pathology, Kangwon National University Hospital, Kangwon National University School of Medicine, Chuncheon-si 24289, Gangwon-do, Korea; pathkyl@kangwon.ac.kr
*   Correspondence: kyuhyoung.lim@kangwon.ac.kr

**Abstract:** The peritoneal carcinomatosis of prostate cancer without bone or other visceral organ involvement is extremely rare. We report a case of an isolated peritoneal metastasis of prostate cancer in a patient without other metastatic sites and a history of prostate surgery. A 63-year-old male with locally advanced prostate cancer without known distant metastasis on androgen deprivation therapy presented with abdominal distension that had persisted for a month. Abdominopelvic computed tomography (CT) showed gastric wall thickening and a moderate amount of ascites. The gastroscopy showed hyperemic mucosal patches on the antrum body. A cytological examination of the ascites fluid was negative for malignant cells. Diagnostic laparoscopy showed multiple nodules in the peritoneum. A biopsy was performed. Histological findings were compatible with metastatic carcinoma of the prostate, which was immunohistochemically positive for pan-cytokeratin, the androgen receptor, and prostate-specific antigen (PSA). The patient was then treated with abiraterone acetate. After 1 month of treatment, both ascites and the PSA value decreased. We describe an extremely rare case of isolated peritoneal carcinomatosis from prostate cancer without any organ metastasis or history of surgery. Clinicians should be aware of these very rare metastases of prostate cancer. Hormonal therapy may be helpful for such cases.

**Keywords:** prostate cancer; peritoneal carcinomatosis; abiraterone acetate

## 1. Introduction

Prostate cancer is the second most common cancer and one of the leading causes of cancer-associated death in men. At the time of diagnosis, approximately 80% of prostate cancer is localized and fully contained within the prostate gland, with a minority of patients having locoregional metastasis (15%) or distant metastasis (5%) [1].

The most common sites of prostate cancer metastasis are locoregional lymph nodes (99%) and bones (84%). Distant lymph nodes (10.6%) and visceral organs such as the liver (~10%), lungs (9.1%), and brain (<2%) are uncommon sites of metastases [2].

Peritoneal carcinomatosis from prostate cancer is very rare, especially when there are no other visceral organ or bone metastases. Thirteen cases of isolated peritoneal carcinomatosis from prostate cancer in patients without a history of surgery have been reported to date. Some cases of peritoneal metastasis after prostate surgery have also been reported [3–5].

Herein, we report a case of an isolated peritoneal carcinomatosis from prostate cancer in a patient without visceral organ or skeletal metastases and no history of previous surgery.

## 2. Case Description

A 61-year-old male patient complained of urinary frequency in September 2019. The patient had an elevated prostate-specific antigen (PSA) value of 60.2 ng/mL. Abdominopelvic computed tomography (CT) showed the multifocal subcapsular extension

of the entire prostate and an invasion of the bladder wall and the right distal ureter. He underwent a transrectal biopsy, and the pathology revealed a Gleason score nine (5 + 4) prostate adenocarcinoma. Whole-body bone scans were negative for skeletal involvement. The locally advanced prostate cancer had been treated with androgen deprivation therapy (ADT), leuprolide, and flutamide since September 2019. During the treatment period, his urinary frequency disappeared, and his PSA level decreased to below the normal range. ADT was maintained for 26 months until November 2021.

In December 2021, the patient was referred to our institute, presenting with gradually worsening abdominal distension that had persisted for a month. His PSA at that time was 586 ng/mL. An abdominal CT showed thickening of the gastric antrum wall and a moderate amount of ascites with peritoneal nodules, suggesting possible gastric cancer with peritoneal metastasis (Figure 1a,b). A gastroscopy showed hyperemic mucosal patches on the antrum body. A histological analysis of the endoscopic biopsy revealed chronic gastritis with intestinal metaplasia. A cytological examination of the ascites was negative for malignant cells. A bone scan showed no evidence of skeletal metastasis.

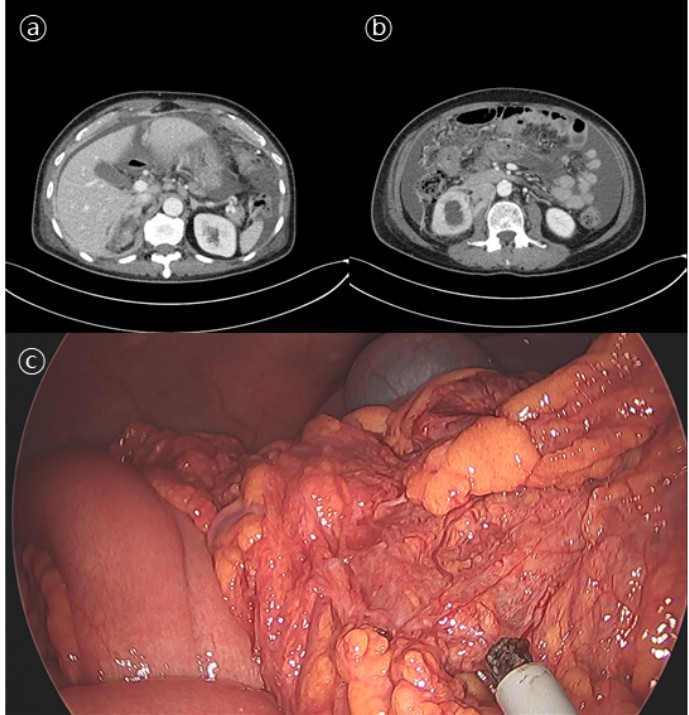

**Figure 1.** Abdominal computed tomography (**a**,**b**) and diagnostic laparoscopic biopsy (**c**) results. (**a**) Abdominal computed tomography showing metastatic lymph nodes in left gastric, retrocaval, aortocaval, and left paraaortic areas. (**b**) Abdominal computed tomography showing massive ascites and peritoneal seeding. (**c**) Diagnostic laparoscopic biopsy showing multiple nodular lesions in the omentum.

The patient underwent diagnostic laparoscopic exploration to determine the cause of the ascites. Multiple nodular lesions in the omentum were observed and biopsied (Figure 1c). As a pathologic finding, tumor cells with a diffuse sheet-like growth pattern were observed. Most tumor cells had a solid growth pattern, with some having a glandular differentiation. Additional immunohistochemical staining was negative for CK7, CK20, TTF-1, and CDX2, whereas staining results for pan-CK, the androgen receptor (AR), AMACR (alpha-methylacyl-CoA racemase), and PSA were positive (Figure 2). These results were consistent with metastatic carcinoma of the prostate.

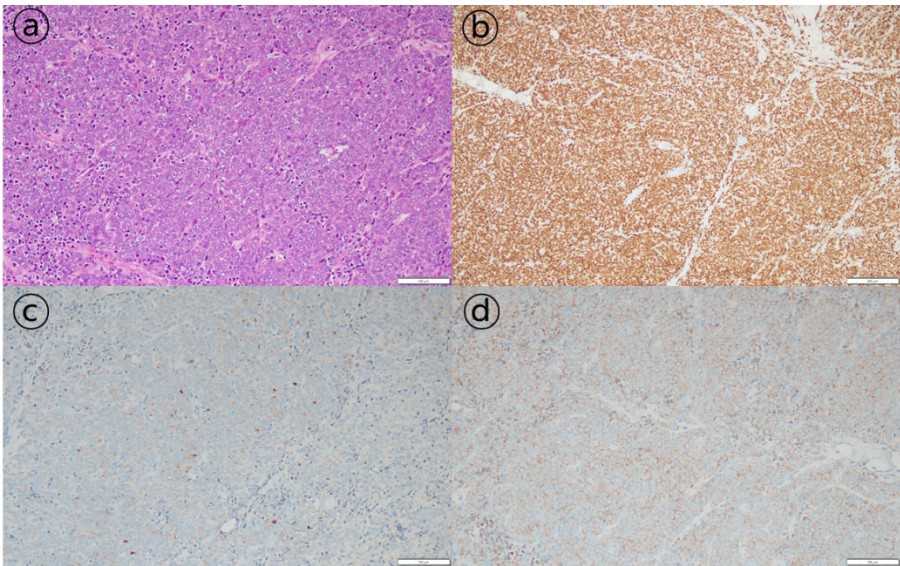

**Figure 2.** Histopathological findings of peritoneal carcinomatosis of prostate cancer. (**a**) Hematoxylin-eosin staining of the peritoneal nodule at a magnification of 200×. (**b**–**d**) Results of immunohistochemistry revealing the tumor was androgen-receptor-positive (magnification 100×), prostate-specific-antigen-positive, and alpha-methylacyl-CoA-racemase-positive (magnification 200×).

The patient started second-line treatment with abiraterone acetate alone in January 2022. One month after the start of abiraterone acetate, ascites significantly decreased, and the PSA level dropped to 1.22 ng/mL (the reference level). The patient is still being followed up with in the outpatient clinic. He continues the treatment with abiraterone acetate.

## 3. Discussion

The peritoneum is a rare metastatic site for prostate cancer. Previously reported studies on peritoneal carcinomatosis of prostate cancer can be categorized into three groups. First, two studies performed a postmortem analysis. Only 5 cases of peritoneal carcinomatosis from prostate cancer were found in 176 postmortem cases [6]. Thirteen isolated peritoneal cases were observed in the autopsy cases of 523 patients with prostate cancer [4]. Second, review studies have shown that peritoneal carcinomatosis of prostate cancer is related to surgical intervention. Approximately 15 reports of iatrogenic peritoneal carcinomatosis associated with prostate cancer surgery were found [4,5]. Third, Delchambre et al. reviewed 13 cases of isolated peritoneal carcinomatosis of prostate cancer without history of surgery [3].

The mechanism of peritoneal prostate cancer metastases is unknown. Based on previously reported studies, iatrogenic peritoneal seeding after surgery can cause rare metastases in patients with prostate cancer. However, cases of isolated peritoneal carcinomatosis without history of previous surgery are presumed to be due to other risk factors of peritoneal metastases. Including our case, 7 of 14 patients had Gleason scores $\geq 9$ at the initial cancer histological analysis [3]. Patient age, initial PSA level, and initial staging at diagnosis have not been shown to be factors predicting isolated peritoneal carcinomatosis in patients with prostate cancer. Further studies such as genomics are necessary to identify the aggressive variants associated with progression to peritoneal carcinomatosis.

The early detection of metastasis in prostate cancer is also important to determine the optimal treatment plan. Including the current patient, a biopsy of metastatic lesion is usually performed to confirm the origin of metastasis. Recently, [68]Ga-prostate-specific membrane antigen (PSMA) PET, a new imaging modality, has received attention because of it being less invasive with higher sensitivity than the conventional modality in patients with prostate cancer. Especially, [68]Ga-PSMA PET has been reported to be more sensitive

than a CT scan in detecting the early lymph node metastasis of prostate cancer, including the peritoneal carcinomatosis of prostate cancer [7,8].

The optimal management for the isolated peritoneal dissemination of prostate cancer has not been established yet. The majority of patients received docetaxel-based chemotherapy. Their overall survival time ranged from 3 weeks to 33 months [3]. Only three patients with isolated peritoneal carcinomatosis were treated with abiraterone acetate. Two of them showed a rapid response as in our case. The other one showed a radiological response for more than 4 years [3,9,10].

Visceral metastases occur mainly in the late stages of cancer. They are correlated with poor outcomes [2]. However, whether patients with isolated peritoneal carcinomatosis have a worse prognosis than those with other visceral metastases of prostate cancer is unknown. Therefore, active attempts to diagnose the cause of atypical presentations of prostate cancer and radical treatment strategies should be considered to control the related symptoms and improve the quality of life.

## 4. Conclusions

We presented a rarely described case of an isolated peritoneal carcinomatosis from prostate cancer in a patient without a previous history of surgery. After treatment with abiraterone acetate, ascites disappeared and the PSA level rapidly decreased within a month. This case highlights the importance of not only a thorough workup process for a very rare presentation of an isolated peritoneal carcinomatosis from prostate cancer, but also a proper therapeutic strategy with abiraterone acetate treatment. Physicians should be aware of this rare case of prostate cancer metastasis. Further studies are needed to identify predictive markers and the optimal treatment for isolated peritoneal carcinomatosis from prostate cancer.

**Author Contributions:** Writing-original draft preparation, H.R.J. and K.-H.L. Writing-review and editing, H.R.J. and K.-H.L. Performed histopathology exams and Writing-review, K.L. All authors have read and agreed to the published version of the manuscript.

**Funding:** This research received no external funding.

**Institutional Review Board Statement:** This study was conducted in accordance with the Declaration of Helsinki. It was approved by the Institutional Review Board of Kangwon National University Hospital (KNUH-2022-05-011).

**Informed Consent Statement:** Informed consent was obtained from the patient for the publication of this case report and related images.

**Data Availability Statement:** The data presented in this study are available on request from the corresponding author. The data are not publicly available due to patient confidentiality.

**Conflicts of Interest:** We have read and understood the current oncology policy on disclosing conflict of interests. We have no conflict of interest relevant to this study to disclose.

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
