# Peer review of "Isolated Peritoneal Metastasis of Prostate Cancer Presenting with Massive Ascites: A Case Report"

_curroncol, doi:10.3390/curroncol29070351_

Round 1

Reviewer 1 Report

Abstract: The introduction part should include that patient had locally advanced prostate cancer without known distant metastasis on ADT. Was the patient also on ADT along with abiraterone?

Line 50: was definitive treatment planned? Or only systemic therapy? Why was ADT stopped in Nov 2021? Need to explain why patient was on ADT only.

Line 70. Was ADT continued while on abiraterone? Did ascites resolve with the therapy?

Please comment on lymph node findings on CT scan. Was there lymphadenopathy?

Author Response

Thank you very much for your comments regarding our manuscript. We are grateful to reviewers for the valuable suggestions provided. Here are our point-by-point responses to the reviewer’s comments.

Our answers are given in blue color. The corresponding changes in the revised manuscript are detectable by “Track Changes”.

Reviewer 1

  1. Abstract: The introduction part should include that patient had locally advanced prostate cancer without known distant metastasis on ADT. Was the patient also on ADT along with abiraterone?

We appreciate your keen comments and completely agree with the reviewer’s comment.

Answer: We corrected our manuscript (“Abstract” , Line 12-14)

“A 63-year-old male with locally advanced prostate cancer without known distant metastasis on androgen deprivation therapy presented with abdominal distension that had persisted for a month.”

The 1st line treatment was leuprolide with flutamide.

After the progression prostate cancer was confirmed, treatment with abiraterone acetate alone was started.

  1. Line 50: was definitive treatment planned? Or only systemic therapy? Why was ADT stopped in Nov 2021? Need to explain why patient was on ADT only.

Thank you for your important comments.

Answer: The patient was initially diagnosed and treated at another hospital. He was referred to our hospital to clarify the cause of massive ascites.

According to the medical records of the hospital where the initial diagnosis and treatment of prostate cancer were performed and the patient’s statements, it is judged that ADT treatment was performed because it was difficult for radical surgery and the patient did not want invasive surgical treatment.

According to the medical records and patient’s statements, as abdominal distension progressed, the patient stopped taking flutamide by his own. Leurpolide was also stopped at the other institution where transferred the patient.

We also revised our manuscript. (“Case description”, Line 50-51)  

“The locally advanced prostate cancer had been treated with androgen deprivation therapy (ADT), leuprolide, and flutamide since September 2019.”

  1. Line 70. Was ADT continued while on abiraterone? Did ascites resolve with the therapy?

We appreciate your important comments.

Answer: After the progression of prostate cancer was confirmed, abiraterone acetate monotherapy was started.

We revised the sentence. (“Case description”, Line 70-71)

“The patient started second-line treatment with abiraterone acetate alone in January 2022.”

  1. Please comment on lymph node findings on CT scan. Was there lymphadenopathy?

We appreciate the reviewer with a helpful recommendation. Your comment reinforces our manuscript.

Answer: We have included CT scan image. (Figure1(a) and Figure 1(b))

“Case description”, Line (77-79)

“(a) Abdominal computed tomography showing metastatic lymph nodes in left gastric, retrocaval, aortocaval, and left paraaortic areas. (b) Abdominal computed tomography showing massive ascites and peritoneal seeding.”

Thank you again for your precious comments.

Reviewer 2 Report

This is a well described case report, and overall informative piece of work which highlights the need for a comprehensive work-up of peritoneal metastasis with confirmation of diagnosis and subsequent change of therapy in case of PSA progression.

It is a very well written paper. The introduction provides a good background to the topic and gives the reader a quick insight into the metastatic pattern of prostate cancer.

I think the motivations for the publication is clear. It is a very rare case of peritoneal metastasis of prostate carcinoma. In particular, after diagnostic confirmation and after the change of therapy, a very good response to therapy with regressive ascites was shown.

In the discussion, the results are clearly highlighted in accordance to other studies.

The authors describe the literature in the context with the results in previous studies and cases, especially in the case of previously not preceded surgery with possible iatrogenic seeding. In addition, the value of the therapy is highlighted. Because of the unknown optimal treatment, the therapy should be more radical. The prognosis in isolated peritoneal carcinomatosis is not known, usually peritoneal metastasis correlates with a worse prognosis. 

Limitations of the study:

CT as an imaging modality may miss potential metastases. Bone scintigraphy was negative for bone metastases. PSMA PET scan, which has higher sensitivity, specificity, and accuracy was not performed or discussed. PSMA PET imaging can detect previously undetected metastases and thus leads to a change in therapy regimen in a certain proportion, this should still be discussed. Example literature I have attached:

Maurer, T., J. E. Gschwend, I. Rauscher, M. Souvatzoglou, B. Haller, G. Weirich, H. J. Wester, M. Heck, H. Kübler, A. J. Beer, M. Schwaiger, and M. Eiber. "Diagnostic Efficacy of (68)Gallium-Psma Positron Emission Tomography Compared to Conventional Imaging for Lymph Node Staging of 130 Consecutive Patients with Intermediate to High Risk Prostate Cancer." J Urol 195, no. 5 (2016): 1436-43.

Roach, P. J., R. Francis, L. Emmett, E. Hsiao, A. Kneebone, G. Hruby, T. Eade, Q. A. Nguyen, B. D. Thompson, T. Cusick, M. McCarthy, C. Tang, B. Ho, P. D. Stricker, and A. M. Scott. "The Impact of (68)Ga-Psma Pet/Ct on Management Intent in Prostate Cancer: Results of an Australian Prospective Multicenter Study." J Nucl Med 59, no. 1 (2018): 82-88.

Was genetic testing performed to identificate germline mutations or alterations in this special case? The impact in personalized therapy should be discussed briefly. 

The following minor comments should still be edited or answered:
1. Please add a comment about the selected patient: was genetic testing performed? 

2. To improve the article, a graphical representation of the findings based on CT images is advisable. 

3. Was there a possibility to perform a PSMA-PET scan to exclude further metastases diagnostically better than CT?

4. Please put the literature references in brackets before the point.

Author Response

Response to Reviewers

17 June 2022

Thank you very much for your comments regarding our manuscript. We are grateful to reviewers for the valuable suggestions provided. Here are our point-by-point responses to the reviewer’s comments.

Our answers are given in blue color. The corresponding changes in the revised manuscript are detectable by “Track Changes”.

  1. Please add a comment about the selected patient: was genetic testing performed?

Thank you for your important comment

Answer: We hope for your generous understanding. Unfortunately, genetic testing was not performed on this patient. We briefly commented on the necessity of genomics to figure out the risk factor predicting isolated peritoneal carcinomatosis of prostate cancer. (“Discussion”, Line 103-104)

“Further studies such as genomics are necessary to identify the aggressive variants associated with progression to peritoneal carcinomatosis.”

  1. To improve the article, a graphical representation of the findings based on CT images is advisable.

Thank you for your keen comments.

Answer: We have added CT scan images. (Figure 1(a) and Figure1(b))

“Case description”, Line (77-79)

“(a) Abdominal computed tomography showing metastatic lymph nodes in left gastric, retrocaval, aortocaval and left paraaortic areas. (b) Abdominal computed tomography showing massive ascites and peritoneal seeding.”

  1. Was there a possibility to perform a PSMA-PET scan to exclude further metastases diagnostically better than CT?

We appreciate the reviewer’s useful suggestion. Your comment reinforces our manuscript.

We carefully read the articles you recommended as examples.

Answer:

During the evaluation, we would have preferred to perform a PSMA PET. Unfortunately, PSMA –PET is not only unavailable in our institution but also our country. We revised the “discussion” section and added references. (Line 106-113)

“The early detection of metastasis in prostate cancer is also important to determine the optimal treatment plan. Including the current patient, biopsy of metastatic lesion is usually performed to confirm the origin of metastasis. Recently, 68Ga–Prostate specific mem-brane antigen (PSMA) PET, a new imaging modality, is getting attention because it is less invasive with higher sensitivity than conventional modality in patients with prostate cancer. Especially, 68Ga-PSMA PET has been reported to be more sensitive in detecting early lymph node metastasis of prostate cancer including peritoneal carcinomatosis of prostate cancer than CT scan [8,9].”

  1. Please put the literature references in brackets before the point.

Thank you for your careful comments.

Answer: We corrected and put the literature references in brackets before the point.

Thank you again for your very important comment.
